# Evaluation of first trimester maternal serum inhibin-A for preeclampsia screening

**Sakita Moungmaithong**[1]☯, **Angel H. Kwan**[2‡], **Ada W. Tse**[2‡], **Natalie K. Wong**[2‡], **Michelle S. Lam**[2‡], **Jing Wang**[3‡], **Liona C. Poon**[2,4]☯*, **Daljit S. Sahota**[2,4]☯*

**1** Department of Obstetrics and Gynaecology, Siriraj Hospital, Mahidol University, Bangkok, Thailand, **2** Department of Obstetrics and Gynaecology, The Chinese University of Hong Kong, Hong Kong SAR, China, **3** Department of Obstetrics and Gynaecology, The First Affiliated Hospital of Sun Yat-Sen University, Guangzhou, China, **4** Shenzhen Research Institute, The Chinese University of Hong Kong, Hong Kong SAR, China

☯ These authors contributed equally to this work.
‡ AHK, AWT, NKW, MSL and JW also contributed equally to this work.
* daljit@cuhk.edu.hk (DSS); liona.poon@cuhk.edu.hk (LCP)

**Data Availability Statement:** All relevant data are within the manuscript and its Supporting Information files.

## Abstract

### Background

International professional organizations recommend aspirin prophylaxis to women screened high risk for preterm preeclampsia (PE) in the first trimester. The UK Fetal Medicine Foundation (FMF) screening test for preterm PE using mean arterial pressure (MAP), uterine artery pulsatility index (UTPI) and placental growth factor (PlGF) was demonstrated to have lower detection rate (DR) in Asian population studies. Additional biomarkers are therefore needed in Asian women to improve screening DRs as a significant proportion of women with preterm and term PE are currently not identified.

### Objectives

To evaluate maternal serum inhibin-A at 11–13 weeks as an alternative to PlGF or as an additional biomarker within the FMF screening test for preterm PE.

### Study design

This is a nested case-control study using pregnancies initially screened at 11–13 weeks for preterm PE using the FMF triple test in a non-intervention study conducted between December 2016 and June 2018. Inhibin-A levels were retrospectively measured in 1,792 singleton pregnancies, 112 (1.7%) with PE matched for time of initial screening with 1,680 unaffected pregnancies. Inhibin-A levels were transformed to multiple of the expected median (MoM). The distribution of $\log_{10}$ inhibin-A MoM in PE and unaffected pregnancies and the association between $\log_{10}$ inhibin-A MoM and gestational age (GA) at delivery in PE were assessed. The screening performance determined by area under receiver operating characteristic curves (AUC) and detection rates (DRs) at a 10% fixed false positive rate (FPR), for preterm and term PE was determined. All risks for preterm and term PE were based on the FMF competing risk model and Bayes theorem. Differences in AUC (ΔAUC) between different biomarker combinations were compared using the Delong test. McNemar's test was used to

**Funding:** This work was supported by i. A start up grant from the Faculty of Medicine, The Chinese University of Hong Kong. ii. Grants from the Health and Medical Research Fund, Hong Kong SAR, China (HMRF-18190821) and iii. The Ministry of Science and Technology (MOST), China (No. 2021YFC2701600). The study sponsors had no role in the study design, collection, analysis, and interpretation of the data, or in the writing of the article.

**Competing interests:** LCP has received speaker fees and consultancy payments from Roche Diagnostics and Ferring Pharmaceuticals. LCP and DS have received in-kind contributions from Roche Diagnostics, PerkinElmer, Thermo Fisher Scientific, and GE Healthcare. This does not alter our adherence to PLOS ONE policies on sharing data and materials. Other co-authors report no conflicts of interest.

assess the off-diagonal change in screening performance at a fixed 10% FPR after adding inhibin-A or replacing PlGF in the preterm PE adjusted risk estimation model.

## Results

Inhibin-A levels in unaffected pregnancies were significantly dependent on GA, maternal age and weight and were lower in parous women with no previous history of PE. Mean $\log_{10}$ inhibin-A MoM in any-onset PE (p<0.001), preterm (p<0.001) and term PE (p = 0.015) pregnancies were all significantly higher than that of unaffected pregnancies. $\log_{10}$ inhibin-A MoM was inversely but not significantly correlated (p = 0.165) with GA at delivery in PE pregnancies. Replacing PlGF with inhibin-A in the FMF triple test reduced AUC and DR from 0.859 and 64.86% to 0.837 and 54.05%, the ΔAUC was not statistically significant. AUC and DR when adding inhibin-A to the FMF triple test were 0.814, 54.05% and the -0.045 reduction in AUC was statistically significant (p = 0.001). At a fixed 10% FPR, replacing PlGF with inhibin-A identified 1 (2.7%) additional pregnancy but missed 5 (13.5%) pregnancies which subsequently developed preterm PE identified by the FMF triple test. Adding inhibin-A missed 4 (10.8%) pregnancies and did not identify any additional pregnancies with preterm PE.

## Conclusion

Replacing PlGF by inhibin-A or adding inhibin-A as an additional biomarker in and to the FMF triple screening test for preterm PE does not improve screening performance and will fail to identify pregnancies that are currently identified by the FMF triple test.

## Introduction

Recent evidence has demonstrated that aspirin prophylaxis can reduce the incidence of pre-term preeclampsia (PE) in asymptomatic pregnant women screened high-risk for preterm PE by 62% [1]. As such, several professional organizations have recommended the first trimester PE screening by the "triple test" developed by the Fetal Medicine Foundation (FMF) for all singleton pregnancies to reduce PE associated maternal and perinatal mortality, morbidity, and their related costs [2–8].

The first trimester triple test consists of maternal factors (MF), measurement of mean arterial pressure (MAP), uterine artery pulsatility index (UTPI), and placental growth factor (PlGF) achieved a superior screening performance for PE compared with the risk factor-based screening [9–12]. Relative differences in detection rates (DRs) of the FMF triple test amongst different racial groups has been reported [9, 10, 12, 13]. At the same 10% false positive rate (FPR), we previously reported that DR in East Asian populations was only 64%, 11% lower than the 75% achieved in mixed European population [9, 13]. The lower DR in Asian women could be explained by the lower incidence of PE risk factors, nevertheless, there may be some other unknown disease modifying factors in which the current test needs to be improved. One way to enhance the performance of the FMF screening test would be the inclusion of additional biomarkers.

Inhibin-A is a glycoprotein expressed during pregnancy by the placenta. Previous studies, assessing inhibin-A after 16 weeks' gestation using enzyme-linked immunoassay (ELISA) assays whilst suitable for explorative research are less useful for routine screening, reported

that maternal serum levels of inhibin-A in pregnancies developing PE are elevated compared to unaffected pregnancies [14–22]. Meta-analysis has shown that aspirin prophylaxis needs to be initiated before 16 weeks to reduce the incidence of preterm PE [23–26]. For inhibin-A to be considered as a potential PE screening marker it would have to be shown to be able to differentiate between PE and non-PE pregnancies before 16 weeks using standard immune-analyzers.

Whilst case-control studies indicate that inhibin-A levels in pregnancies with established PE are significantly higher [17, 19, 27–29], there still remains a lack of consensus as to which temporal timepoint the difference in levels is evident; the additional improvement in DR that adding inhibin-A would result in when screening for preterm and term PE as opposed to early (< 34 weeks) and late (≥34 weeks) onset PE; whether differences reported in earlier studies based on measurement of inhibin-A by ELISA are comparable to immunoassay measurements. The objectives of this study were firstly to determine whether inhibin-A levels in PE pregnancies determined using a standard immunoassay were increased at 11–13 weeks and secondly if so whether its use would improve the DRs for preterm PE if incorporated within the FMF triple test.

## Materials and methods

This was a nested case-control study using archived sera collected from 6,546 consecutive Chinese women with a singleton pregnancy, which resulted in a livebirth from a prospective cohort of pregnant women at 11–13 weeks' gestation attending Down Syndrome screening program at our institution between December 2016 and June 2018 and participated in a Asian-based study for the validation of the FMF triple test preterm PE screening [13]. All eligible women who agreed to participate were asked to provide written informed consent for storage and use of archived sera in future research at the time of screening. Ethical approval for the base-cohort study was obtained from the Joint Chinese University of Hong Kong—New Territories East Cluster Clinical Research Ethics Committee (CREC Ref. No. 2016.152). Excess serum was stored at -80˚C for future research. Authors had access to information that could identify individual participants after data collection. The ethical approval for this retrospective study (CREC Ref. No.: 2021.258) was obtained from the Joint Chinese University of Hong Kong—New Territories East Cluster Clinical Research Ethics Committee. Data on pregnancy outcomes were collected from the hospital maternity electronic records. The diagnosis of PE was based on the definition of the International Society for the Study of Hypertension in Pregnancy [4]. Preterm and term PE was defined as delivery with PE at <37 and ≥37 weeks' gestation, respectively. This work was supported by i. A start up grant from the Faculty of Medicine, The Chinese University of Hong Kong. ii. Grants from the Health and Medical Research Fund, Hong Kong SAR, China (HMRF-18190821) and iii. The Ministry of Science and Technology (MOST), China (No. 2021YFC2701600). The study sponsors had no role in the study design, collection, analysis, and interpretation of the data, or in the writing of the article.

The primary aims of the study were to determine if the addition of inhibin-A would increase the DR and screening accuracy of the FMF triple test. Our previous studies indicated that before adjusting for aspirin use the DR and areas under the receiver operational curves (AUC) when screening for preterm PE using the FMF triple test were 59% and 0.85 respectively [30], whilst the DR and AUC when screening for any onset PE using maternal history, MAP and PlGF were 51% and 0.79, respectively [31].

The sample size of the present study was therefore based on the assumption that the addition of inhibin-A to the screening test for any onset PE would detect an improvement in AUC

from 0.8 to 0.85, a ΔAUC = 0.05. Assuming a correlation between screening tests in those screened high risk and low risk of 0.8 it would require 107 pregnancies with PE and a ratio between non-PE and PE pregnancies of 15 to achieve a power of 80% for a type 1 error of 5%. In the base-cohort study reported by Wah *et al.*, 112 (1.7%) women developed PE ("PE"), 37 (0.6%) and 75 (1.1%) of whom had preterm and term PE [30]. Fifteen unaffected pregnancies screened within ± 30 days of each PE affected pregnancy were therefore randomly selected from amongst the available screened pregnancies who did not develop PE to form an unaffected group for comparison of PE screening performance.

## Measurement of inhibin-A

Archived sera were retrieved from -80˚C storage and thawed in batches using a slow defrost protocol. Inhibin-A levels were determined using the BRAHMS KRYPTOR Gold Analyzer (ThermoFisher Scientific, Hennigsdorf, Germany). The inter- and intra-assay coefficient of variation of the analyzer inhibin-A immunoassay ranged from 2–6%.

## Adjusting inhibin-A levels for covariate fixed and random factors

Inhibin-A levels of women with unaffected pregnancy were transformed to their equivalent $\log_{10}$ values then assessed to determine whether their levels were independent of gestational age (GA) at time of blood draw, maternal age, weight, height, parity and history of PE in the previous pregnancy (nulliparous, parous with prior PE, parous without prior PE), smoking status, diabetic status, and method of conception (spontaneous, IVF [*in vitro* fertilization]) using univariate and multivariate regression analyses. Fixed and random factors with a significant impact ($p < 0.05$) on $\log_{10}$ inhibin-A level were retained in the final model used to calculate the expected median values of $\log_{10}$ inhibin-A. Inhibin-A levels were then transformed to their equivalent multiple of the expected median (MoM) values. Gestational age and weight were centered on 77 days and 69 kg, respectively, the same values as those previously used by Tan *et al.* [32].

## Estimation of preterm preeclampsia risk

Individual women's *a priori* risks of delivery with preterm PE and term PE were estimated based on maternal factors (MF) using the FMF competing risk model [9]. The *posteriori* risks were determined using Bayes theorem-based approach by adjusting the *a priori* with the likelihood function of biomarker measurements.

All risks for PE were estimated using our in-house laboratory PE risk calculation software and MAP, UTPI and PlGF were transformed to their equivalent MoM values using published transformation models [32, 33]. Expected levels and biomarker MoM distributions of MAP, UTPI and PlGF in unaffected and PE pregnancies used to estimate the *posteriori* risks were previously reported in the National Institute for Health Research SPREE study [34].

Our earlier studies indicated that the overall median PlGF MoM in non-PE pregnancies in East Asians using the FMF PlGF MoM transformation model gave a distribution with a median of 0.84 [35]. Prior to calculating the PE risks, PlGF MoMs were corrected to recenter the PlGF MoM distribution to a median of 1 MoM in order to avoid overestimation of preterm PE risks. Gestational age at the time of screening used to derive the biomarker MoM values was determined from the fetal CRL measurement using a previously published Chinese dating formula [36].

## Data processing and statistical analysis

Maternal demographic and biomarker characteristics were presented in median (interquartile range [IQR]). Comparisons were performed by Mann-Whitney U test for continuous variables

and Chi-square test for categorical variables. Descriptive statistics was used to describe the central tendency and distribution of $\log_{10}$ inhibin-A MoM in any onset PE, preterm PE, term PE and unaffected pregnancies. Independent samples t-test and ANOVA with Bonferroni correction for multiple comparison (p value<0.016) were used to determine the differences in mean levels of $\log_{10}$ inhibin-A MoM between outcome groups. Correlation between $\log_{10}$ inhibin-A MoM and other biomarkers $\log_{10}$ transformed MoM values in PE, unaffected as well as in all pregnancies was assessed. Linear regression analysis was performed to determine whether the level of $\log_{10}$ inhibin-A MoM at 11–13 weeks was associated with gestational age at delivery in those with PE.

### Comparison of for preterm and term preeclampsia screening performance

Receiver operating characteristic (ROC) curves were constructed and AUCs were determined [37]. DRs at a 10% fixed FPR and corresponding risk level (1:XXX) were determined for each of the different screening biomarker combinations assessed. Differences between AUCs were tested for significance using the Delong test [38]. McNemar's test was used to determine the percentage change in the off-diagonal probabilities by adding or removing inhibin-A to the PE screening test biomarker at a fixed 10% FPR.

Statistical Product and Service Solutions (SPSS) for Windows version 20 (SPSS, Illinois, USA) and MedCalc Statistical Software version 18.10.2 (MedCalc Software bvba, Ostend, Belgium; http://www.medcalc.org; 2018) were used for statistical analyses. Tests were considered statistically significant if p-value <0.05.

## Results

The maternal, pregnancy and screening biomarker characteristics according to pregnancy outcomes are summarized in Table 1. Women who developed PE showed the expected traits with regard to obstetric and medical history as well as levels of their biomarkers. Compared with the unaffected pregnancies, women with PE had higher maternal age and BMI, higher rates of IVF conception, nulliparity and previous history of PE, and lower rate of history of previous pregnancy without PE. There was no difference in the rates of smoking, chronic hypertension and SLE/APS between groups. Although there was a higher rate of pre-existing diabetes mellitus in the PE group, compared to the unaffected group, the total number of cases was small. Inhibin-A levels at 11–13 weeks in non-PE pregnancies were found to be significantly dependent on GA (p<0.001), maternal age (p<0.001), maternal weight (Wgt) (p<0.001) and parous without prior PE (p<0.001). The final model used to transform measured inhibin-A to it equivalent MoM value is reported in S1 Table.

The distribution of $\log_{10}$ inhibin-A MoM in unaffected pregnancies was Gaussian with a mean of -0.0135 and a standard deviation (SD) of ±0.2108. Mean (SD) $\log_{10}$ MoM inhibin-A in any-onset PE, preterm PE and term PE pregnancies were significantly higher than that of unaffected pregnancies [0.0911 (±0.2571), 0.1593 (±0.2657), 0.0575 (±0.2477) with p-values of <0.001, <0.001 and 0.015, respectively]. S2 Table summarizes the observed covariance and correlation coefficients of inhibin-A with that of existing biomarkers. Levels of $\log_{10}$ inhibin-A MoM in pregnancies that developed PE were negatively associated with GA at delivery (S1 Fig). The correlation however failed to reach statistical significance (r = -0.132, p = 0.165).

### Preterm preeclampsia prediction

For the prediction of preterm PE, the AUC when screening with MF only was 0.719 (95% confidence interval [CI], 0.63–0.81). Only screening by MF plus PlGF significantly increased the AUC when compared with MF only (p = 0.021). There were no significant differences among

**Table 1. Summary of maternal demographic and biomarker characteristics.**

| Characteristics | Unaffected (n = 1,680) | Any-onset PE (n = 112) | Preterm PE (n = 37) | Term PE (n = 75) |
|---|---|---|---|---|
| Maternal Age at EDD | 32.49 (29.37–35.37) | 33.97 (30.18–36.55) * | 34.50 (30.25–36.72) | 33.51 (30.02–36.39) |
| Maternal Weight (Kg) | 53.40 (49.10–59.90) | 54.85 (49.53–63.35) | 53.50 (49.45–50.80) | 56.60 (49.40–65.50) |
| Maternal Height (cm) | 158.00 (154.00–162.00) | 156.00 (152.00–159.00) * | 155.00 (151.00–158.50) † | 156.00 (152.00–160.00) |
| Body Mass Index (Kg/m$^2$) | 21.43 (19.81–23.82) | 22.96 (20.67–25.47) | 21.98 (20.72–24.29) | 23.89 (20.49–26.51) ‡ |
| In-vitro fertilization | 29 (1.73) | 8 (7.14) * | 0 (0.00) | 8 (10.67) |
| Smoker | 125 (7.44) | 7 (6.25) | 1 (2.70) | 6 (8.00) |
| Chronic HT | 6 (0.36) | 1 (0.89) | 1 (2.70) | 0 (0.00) |
| SLE/APS | 2 (0.12) | 0 (0.00) | 0 (0.00) | 0 (0.00) |
| Type I or II DM | 3 (0.18) | 2 (1.79) * | 1 (2.70) | 1 (1.33) |
| Parity | | | | |
| Nulliparous | 822 (48.93) | 79 (70.54) * | 24 (64.86) | 55 (73.33) |
| Parous—No prior PE | 850 (50.60) | 29 (25.89) * | 11 (29.73) | 18 (24.00) |
| Parous—Prior PE | 8 (0.48) | 4 (3.57) * | 2 (5.41) | 2 (2.67) |
| Family History of PE | 10 (0.60) | 2 (1.79) | 1 (2.70) | 1 (1.33) |
| CRL (mm) | 59.65 (55.30–64.10) | 57.95 (55.05–63.31) | 56.90 (52.70–62.55) | 58.70 (56.10-63-60) |
| GA at assessment (days) | 87.00 (85.00–90.00) | 86.00 (85.00–89.00) | 86.00 (8300–88.50) | 87.00 (85.00–89.00) |
| Measured and standardized biomarker levels | | | | |
| MAP MoM | 0.97 (0.92–1.04) | 1.03 (0.96–1.14) * | 1.05 (0.96–1.16) † | 1.02 (0.96–1.25) ‡ |
| UTPI MoM | 1.07 (0.91–1.25) | 1.14 (0.94–1.34) * | 1.21 (1.04–1.43) † | 1.11 (0.92–1.27) |
| PlGF MoM | 1.01 (0.75–1.32) | 0.76 (0.48–1.04) * | 0.58 (0.41–0.83) † | 0.82 (0.62–1.08) ‡ |

Summary of maternal demographic and biomarker characteristics of pregnant women with measured inhibin-A concentration levels at 11 to 13 weeks' gestation according to pregnancy outcome status.

Data are summarized as median (interquartile range) or number (%) (* p-value <0.05 between Unaffected vs. any-PE, † p-value <0.016 between Unaffected vs. Preterm PE, ‡ p-value <0.016 between Unaffected vs. Term PE)

various two-biomarker combinations as well as among three-biomarker combinations. Tables 2 and 3 and Fig 1 summarize the overall screening performance of various combinations of biomarkers including adding inhibin-A or replacing PlGF with inhibin-A for the prediction of preterm PE. The best AUC was achieved by MF + (MAP, PlGF, inhibin-A) at 0.861 (95%CI, 0.802–0.919). Screening by MF + (MAP, UTPI, inhibin-A) had significantly better

**Table 2. Preterm preeclampsia screening performance.**

| Screening Test Biomarker Combination | Area Under Curve (95% CI) | 10% False positive rate | |
|---|---|---|---|
| | | Risk Cut-off (1: XXX) | Detection Rate (% [95% CI]) |
| MF alone | 0.719 (0.628–0.810) | 1:138 | 37.84% (24.32–56.76%) |
| MF + PlGF + inhibin-A | 0.835 (0.769–0.900) | 1:117 | 54.05% (40.54–72.97%) |
| MF + MAP + UTPI + inhibin-A | 0.837 (0.766–0.908) | 1:131 | 54.05% (40.54–72.97%) |
| MF + MAP + UTPI + PlGF | 0.859 (0.792–0.927) | 1:126 | 64.86% (48.65–81.08%) |
| MF + MAP + PlGF + inhibin-A | 0.861 (0.802–0.919) | 1:139 | 59.46% (43.24–75.68%) |
| MF + MAP + UTPI + PlGF + inhibin-A | 0.814 (0.740–0.889) | 1:110 | 54.05% (40.54–72.97%) |

Preterm preeclampsia screening performance using different combinations of maternal factors (MF), mean arterial pressure (MAP), uterine artery pulsatility index (UTPI), placental growth factor (PlGF) and inhibin-A

**Table 3. Comparison of overall screening performance for preterm preeclampsia.**

| Screening test biomarker combination | Area under Curve | | p-value |
|---|---|---|---|
| | ΔAUC | 95% CI | |
| *Compared with MF only* | | | |
| MF + PlGF + inhibin-A | 0.115 | 0.039 to 0.191 | 0.003* |
| MF + MAP + UTPI + inhibin-A | 0.116 | 0.051 to 0.182 | <0.001* |
| MF + MAP + UTPI + PlGF | 0.138 | 0.059 to 0.218 | 0.001* |
| MF + MAP + UTPI + PlGF+ inhibin-A | 0.095 | 0.005 to 0.185 | 0.038* |
| *Compared with MF + PlGF + inhibin-A* | | | |
| MF + MAP + UTPI + inhibin-A | 0.001 | -0.054 to 0.057 | 0.965 |
| MF + MAP + UTPI + PlGF | 0.023 | -0.013 to 0.060 | 0.210 |
| MF + MAP + UTPI + PlGF+ inhibin-A | -0.020 | -0.071 to 0.032 | 0.450 |
| *Compared with MF + MAP + UTPI + PlGF* | | | |
| MF + (MAP, PlGF, Inhibin-A) | 0.001 | -0.030 to 0.032 | 0.935 |
| MF + MAP + UTPI + inhibin-A | -0.022 | -0.074 to 0.030 | 0.402 |
| MF + MAP + UTPI + PlGF + inhibin-A | -0.045 | -0.073 to -0.017 | 0.001* |
| *Compared with MF + MAP + UTPI + inhibin-A* | | | |
| MF + MAP+ UTPI + PlGF+ inhibin-A | -0.021 | -0.096 to 0.054 | 0.580 |

Comparison of overall screening performance for preterm preeclampsia using the different combinations of maternal factors (MF), mean arterial pressure (MAP), uterine artery pulsatility index (UTPI), placental growth factor (PlGF) and inhibin-A

performance than screening by MF alone and any two-biomarker combinations. The screening performance using MF + (MAP, UTPI, inhibin-A) achieved lower AUC than MF + (MAP, UTPI, PlGF) but did not reach statistical significance. Screening by MF + (MAP, UTPI, PlGF,

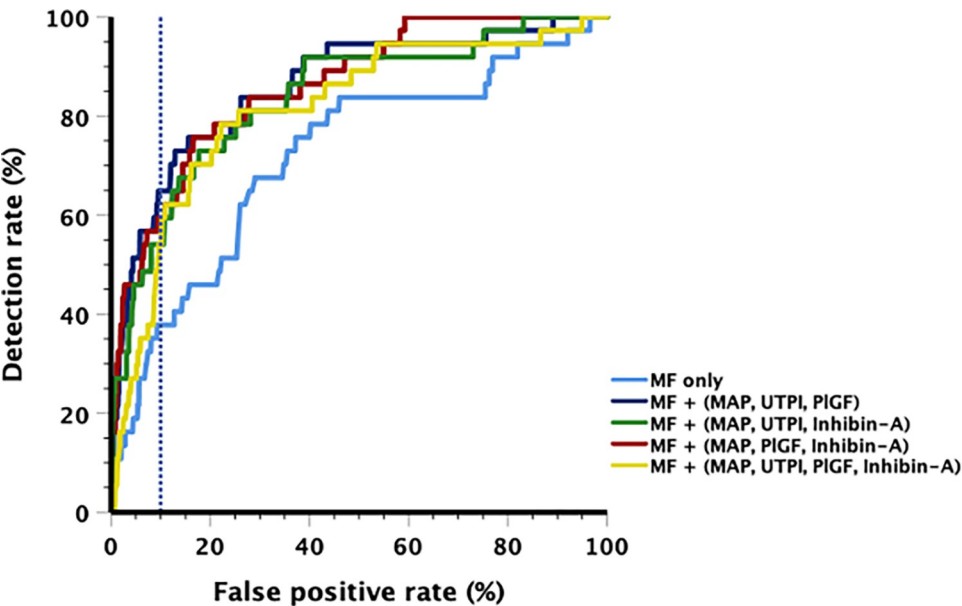

**Fig 1. Comparative screening performance for preterm preeclampsia.** The screening performance for preterm preeclampsia when replacing or combining placental growth factor with inhibin-A in conjunction with maternal factors, mean arterial pressure and uterine artery (Dashed vertical line represent 10% false positive rate).

**Table 4. Term preeclampsia screening performance.**

| Screening Test Biomarker Combination | Area under Curve (95% CI) | 10% False positive rate | |
|---|---|---|---|
| | | Risk Cut-off (1: XXX) | Detection Rate (% 95% CI) |
| MF alone | 0.719 (0.663–0.776) | 1:28 | 37.33% (26.13–48.54%) |
| MF + PlGF + Inhibin-A | 0.757 (0.700–0.814) | 1:25 | 42.67 (31.21–54.12%) |
| MF + MAP + UTPI + Inhibin-A | 0.769 (0.717–0.821) | 1:24 | 38.67 (27.39–49.95%) |
| MF + MAP + UTPI + PlGF | 0.785 (0.734–0.836) | 1:24 | 44.00 (32.50–55.50%) |
| MF + MAP + PlGF + Inhibin-A | 0.791 (0.740–0.841) | 1:24 | 38.67 (27.39–49.95%) |
| MF + MAP + UTPI + PlGF + Inhibin-A | 0.761 (0.710–0.812) | 1:22 | 37.33 (26.13–48.54%) |

Term preeclampsia screening performance using different combinations of maternal factors (MF), mean arterial pressure (MAP), uterine artery pulsatility index (UTPI), placental growth factor (PlGF) and inhibin-A

inhibin-A) significantly worsened the screening performance as compared with screening by MF + (MAP, UTPI, PlGF) ($\Delta$AUC = -0.045, p = 0.001). At a fixed 10% FPR, a DR for preterm PE using MF only was 37.84% (95%CI, 24.32–56.76%) and screening by MF + (MAP, UTPI, PlGF) yielded the best DR at 64.86% (95%CI, 48.65–81.08%).

McNemar's test indicated that substituting PlGF with inhibin-A in combination with MAP and UTPI identified only 1 (2.7%) additional pregnancy but missed 5 (13.5%) pregnancies developing preterm PE that could have been identified by PlGF. Combining inhibin-A with maternal factors and the existing FMF triple test missed 4 (10.8%) pregnancies and did not identify any additional pregnancies developing preterm PE.

## Term preeclampsia prediction

For the prediction of term PE, Tables 4 and 5 and Fig 2 summarize the overall screening performance of various combinations of biomarkers. Compared with the performance of MF alone for term PE prediction, which achieved a similar AUC as preterm PE prediction at 0.719 (95%CI, 0.66–0.78), only a combination of MF with PlGF could improve the screening performance ($\Delta$AUC = 0.035, p = 0.037). Substituting PlGF with inhibin-A in the FMF triple test performed better than MF alone ($\Delta$AUC = 0.050, p = 0.038) but did not improve screening performance of term PE prediction compared with the FMF triple test ($\Delta$AUC = -0.016, p = 0.152). Combining inhibin-A to the FMF triple test did not yield any improvement in term PE screening performance over MF only ($\Delta$AUC = 0.041, 0.098) and MF + (MAP, UTPI, inhibin-A) ($\Delta$AUC = -0.008, p = 0.611), whilst it demonstrated inferior performance compared with the Triple test ($\Delta$AUC = -0.024, <0.001).

McNemar's test for term PE prediction demonstrated that substituting PlGF with inhibin-A in combination with MAP and UTPI identified 4 (5.3%) additional pregnancies but missed 10 (13.3%) pregnancies developing term PE that could have been identified by PlGF. Combining inhibin-A with maternal factors and the existing FMF triple test missed 6 (8%) pregnancies and identified 1 (1.3%) additional pregnancy with term PE.

## Discussion

### Principal findings

Our study has demonstrated that, although levels of inhibin-A at 11–13 weeks' gestation are significantly higher in pregnancies that subsequently develop PE compared with unaffected

**Table 5. Comparison of overall screening performance for term pre-eclampsia.**

| Screening test biomarker combination | Area under Curve | | p-value |
|---|---|---|---|
| | ΔAUC | 95%CI | |
| *Compared with MF only* | | | |
| MF + (PlGF, Inhibin-A) | 0.038 | 0.001 to 0.074 | 0.043* |
| MF + (MAP, UTPI, Inhibin-A) | 0.050 | 0.003 to 0.096 | 0.038* |
| MF + (MAP, UTPI, PlGF) | 0.065 | 0.018 to 0.113 | 0.007* |
| MF + (MAP, UTPI, PlGF, Inhibin-A) | 0.041 | -0.008 to 0.090 | 0.098 |
| *Compared with MF + (PlGF, Inhibin-A)* | | | |
| MF + (MAP, UTPI, Inhibin-A) | 0.012 | -0.034 to 0.058 | 0.613 |
| MF + (MAP, UTPI, PlGF) | 0.028 | -0.016 to 0.072 | 0.218 |
| MF + (MAP, UTPI, PlGF, Inhibin-A) | 0.004 | -0.046 to 0.053 | 0.883 |
| *Compared with MF + (MAP, UTPI, PlGF)* | | | |
| MF + (MAP, PlGF, Inhibin-A) | 0.006 | -0.009 to 0.021 | 0.461 |
| MF + (MAP, UTPI, Inhibin-A) | -0.016 | -0.038 to 0.006 | 0.152 |
| MF + (MAP, UTPI, PlGF, Inhibin-A) | -0.024 | -0.037 to 0.012 | <0.001* |
| *Compared with MF + MAP + UTPI + Inhibin-A* | | | |
| MF + (MAP, UTPI, PlGF, Inhibin-A) | -0.008 | -0.040 to 0.023 | 0.611 |

Comparison of overall screening performance for term pre-eclampsia using the different combinations of maternal factors (MF), mean arterial pressure (MAP), uterine artery pulsatility index (UTPI), placental growth factor (PlGF) and inhibin-A

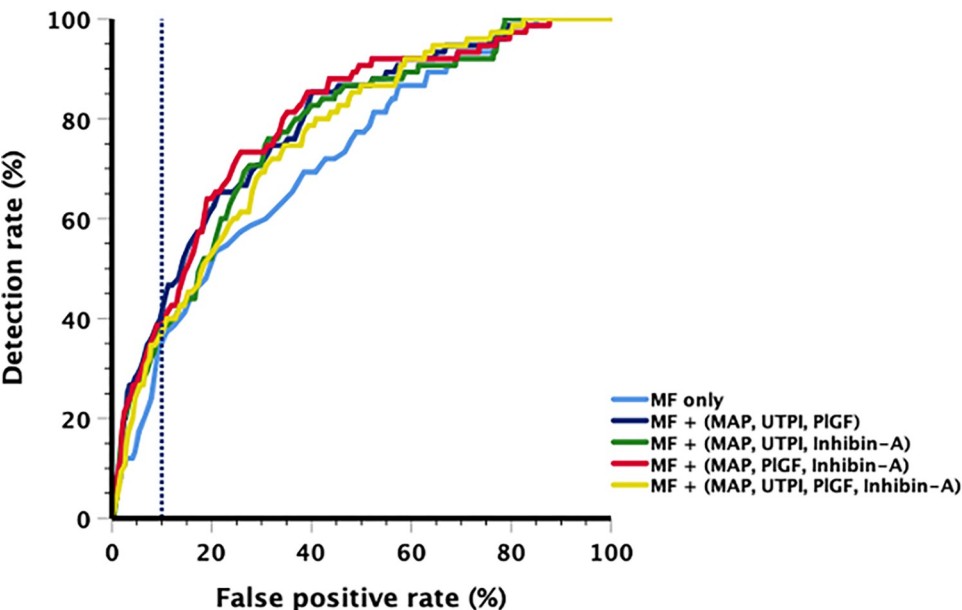

**Fig 2. Comparative screening performance for term preeclampsia.** The screening performance for term preeclampsia when replacing or combining placental growth factor (PlGF) with inhibin-A in conjunction with maternal factors (MF), mean arterial pressure (MAP) and uterine artery (UTPI) (Dashed vertical line represent 10% false positive rate).

pregnancies, replacing PlGF with inhibin-A in the existing FMF triple test does not improve the performance of preterm and term PE screening; while adding inhibin-A as the fourth biomarker decreases the screening performance.

## Results

Inhibin-A levels in pregnancies with established PE have been reported as being significantly higher than those in unaffected pregnancies [17, 19, 27–29] leading to the postulation that increased levels of inhibin-A in PE pregnancies is a consequence of trophoblast dysfunction allowing potential prediction of when and in which pregnancy PE could occur [33, 39–41].

Muttukrishna *et al.* conducted a longitudinal study and indicated that significant differences in inhibin-A levels, determined by ELISA, were only evident after 15–19 weeks of gestation in those who developed early onset PE [19]. The authors however did not correct for independent factors such as GA and maternal weight, both of which would have been expected to impact on the determined inhibin-A level. In contrast, Akolekar *et al.*, Spencer *et al.* and Poon *et al.* [14, 42, 43], all using an ELISA test, reported significantly increased levels of inhibin-A at 11–13 weeks in plasma and serum after correcting for maternal and pregnancy factors. Poon *et al.* in a later study reconfirmed that inhibin-A levels were increased at 11–13 weeks in PE affected pregnancies using a standard immunoassay for the measurement of inhibin-A after correcting for maternal and pregnancy factors [34]. Our data and analysis confirmed that inhibin-A levels determined by immunoassay at 11–13 weeks are similarly increased in PE affected pregnancies of East Asian women indicating that inhibin-A could be used for PE screening. We have further demonstrated that there are lower inhibin-A levels in parous women without prior PE, which reduces the risk for developing PE in the current pregnancy. This finding may indicate that women with a previous unaffected pregnancy have the ability to accomplish adequate maternal adaptation to tolerate pregnancy-related stress in the previous pregnancy, thus providing a protective effect against the development of PE in the current pregnancy.

Using the gestational 90th percentile of inhibin-A in normotensive pregnancies as a cut-off, Muttukrishna *et al.* reported screening sensitivities at 15–19 weeks for any and early onset PE of 28% and 67%, respectively, for a 12% FPR [20]. Using a similar cut-off, Spencer *et al.* reported that inhibin-A had a DR of 35% for a 5% FPR which increased to 67% for the same FPR when combined with second trimester UTPI [43]. Akolekar *et al.* reported similar DRs of 23% and 31% when screening by inhibin-A alone for 5% and 10% FPR and that DR increased to 85% and 89% at the same FPRs when combined with maternal history and UTPI [14]. However, when screening for early onset PE in combination with PlGF and inhibin-A, Poon *et al.* reported that only PlGF and not inhibin-A remained as an independent predictor for development of early onset PE when constructing a multivariate logistic regression prediction model for early onset PE [42]. The current approach to screening for PE has evolved from to one based on estimating marginal probability of an event in the presence of competing event combined with Bayes theorem allowing multiple biomarkers to be combined and assessment of screening performance. Using this approach Poon *et al.* in a recent study showed that inhibin-A did not improve the DRs of any-onset PE and preterm PE over the prediction provided by PlGF in a predominantly White, Black and South Asian population [34]. Our analysis and findings in East Asian women would confirm Poon *et al.* study with regard to concurrent use of inhibin-A and PlGF to screen for preterm PE at 11–13 weeks.

## Clinical implications

This study demonstrating that adding inhibin-A to or replacing a biomarker by inhibin-A in the currently used FMF triple test does not significantly improve the screening performance

could be explained by the concept of diminishing marginal returns and the fact that inhibin-A levels significantly correlate with PlGF levels in the PE group (S2 Table).

## Research implications

Our findings in this present study highlight that although a potential biomarker is associated with development of PE its inclusion in a risk estimation model for development of PE will not necessarily improve screening performance if it is significantly correlated with existing biomarkers or if its discriminatory performance, reflected by Mahalanobis distance, is on a par with existing biomarkers. Mahalanobis distance of inhibin-A in this study was 0.82 for preterm PE which is similar to the 0.88 estimated by Cuckle for early-onset PE [44]. The MAP, UTPI and PlGF biomarker combination to screen for PE remains the best combination to use in the first trimester, and that PlGF will not be replaced by another placentally derived biomarker unless and until it has significantly higher Mahalanobis distance which exceeds that of PlGF and similar to that of MAP and UTPI or alternatively that any new biomarker incorporated reflects a different pathogenesis or dimension of the PE, such as maternal endothelial, cardiac and end-organ dysfunction related biomarkers [45, 46].

## Strengths and limitations

The strengths of our study were firstly, that inhibin-A was measured using a readily available immunoassay and immune-analyzer platform; secondly, that we assessed screening performance using the competing risk and Bayes based model, a de facto gold standard approach; and lastly that we assessed relative improvement or loss of screening performance on an individual case basis. A limitation of the current study was that we did not adjust the screening performance for aspirin prophylaxis.

## Conclusions

In conclusion, inhibin-A is significantly elevated at 11–13 weeks' gestation in pregnancies that subsequently develop PE but has lower predictive performance for the screening of both preterm and term PE in the first trimester when compared with PlGF. Neither replacing PlGF by inhibin-A nor adding inhibin-A could enhance the screening performance of the FMF triple test for both preterm and term PE.

## Supporting information

**S1 Fig. Association between $\log_{10}$ inhibin-A multiple of median (log10 inhibin-A MoM) and gestation at delivery in women who developed preeclampsia.** Dashed lines represent the observed expected mean level of $\log_{10}$ inhibin-A MoM and its 95% upper and lower confidence interval.
(TIF)

**S1 Table. Model derived from 1680 unaffected pregnancies used to estimate the expected ThermoFisher BRAHMS KRYPTOR $\log_{10}$ inhibin-A in at 11–13 weeks.**
(DOCX)

**S2 Table. Observed $\log_{10}$ multiple of the median biomarker standard deviation and inter–biomarker correlations in women with and without preeclampsia.**
(DOCX)

## Acknowledgments

We wish to thank the members of the Obstetrics Screening Laboratory, Maternal Fetal Medicine team, midwives, nurses, research students and assistants at the Prince of Wales Hospital in facilitating the performance of this study.

## Author Contributions

**Conceptualization:** Natalie K. Wong, Liona C. Poon, Daljit S. Sahota.

**Data curation:** Angel H. Kwan, Ada W. Tse, Michelle S. Lam, Jing Wang.

**Formal analysis:** Sakita Moungmaithong, Liona C. Poon, Daljit S. Sahota.

**Funding acquisition:** Daljit S. Sahota.

**Investigation:** Sakita Moungmaithong.

**Methodology:** Liona C. Poon, Daljit S. Sahota.

**Software:** Daljit S. Sahota.

**Supervision:** Liona C. Poon, Daljit S. Sahota.

**Validation:** Sakita Moungmaithong, Daljit S. Sahota.

**Visualization:** Sakita Moungmaithong.

**Writing – original draft:** Sakita Moungmaithong.

**Writing – review & editing:** Sakita Moungmaithong, Liona C. Poon, Daljit S. Sahota.

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
