## [Decision Letter · Decision Letter 0]

27 Mar 2023

PONE-D-22-29184Evaluation of first trimester maternal serum inhibin-A for preeclampsia screeningPLOS ONE

Dear Dr. Moungmaithong,

Thank you for submitting your manuscript to PLOS ONE. After careful consideration, the manuscript received favorable reviews and requires minor revision to address the concerns raised by the reviewers. Therefore, we invite you to submit a revised version of the manuscript that addresses the points raised during the review process.

We look forward to receiving your revised manuscript.

Kind regards,

Offer Erez, M.D.

Academic Editor

PLOS ONE

Journal Requirements:

2. Please amend your ethics statement in the Methods section of your manuscript. Specifically, please state that the ethics approval for this retrospective study (CREC Ref. No.: 2021.258) was obtained from the Joint Chinese University of Hong Kong - New Territories East Cluster Clinical Research Ethics Committee.

This work was supported by a grant from the Health and Medical Research Fund, Hong Kong SAR, China (HMRF-18190821). The study sponsor had no role in the study design, collection, analysis, and interpretation of the data; or in the writing of the article. The base-cohort study “Prospective validation of prediction algorithms for preeclampsia in the first-, second- and third-trimesters of pregnancy” was supported by a grant from Department of Obstetrics and Gynaecology, Prince of Wales Hospital, The Chinese University of Hong Kong (CUHK), Hong Kong SAR.

LCP has received speaker fees and consultancy payments from Roche Diagnostics and Ferring Pharmaceuticals. LCP and DS have received in‐kind contributions from Roche Diagnostics, PerkinElmer, Thermo Fisher Scientific, and GE Healthcare. Other co-authors report no conflicts of interest.

5. Please amend your manuscript to include your abstract after the title page.

6. Please include your tables as part of your main manuscript and remove the individual files. Please note that supplementary tables (should remain/ be uploaded) as separate "supporting information" files.

Reviewers' comments:

Reviewer's Responses to Questions

**Comments to the Author**

1. Is the manuscript technically sound, and do the data support the conclusions?

Reviewer #1: Yes

Reviewer #2: Yes

2. Has the statistical analysis been performed appropriately and rigorously? 

Reviewer #1: Yes

Reviewer #2: Yes

3. Have the authors made all data underlying the findings in their manuscript fully available?

Reviewer #1: Yes

Reviewer #2: Yes

4. Is the manuscript presented in an intelligible fashion and written in standard English?

Reviewer #1: Yes

Reviewer #2: Yes

5. Review Comments to the Author

Reviewer #1: Not withstanding its acknowledged low dose aspirin issue limitation and retrospective design, this is a clearly described, statistically sound and biomedically important contribution to the current literature dealing with early pregnancy biomarkers that may help predict preeclampsia later in a pregnancy.

Although the conclusion of the study is in one sense "negative" in so far as the study results do not support the use of inhibin as a biomarker in place of or in addition to currently used options (such as PlGF in particular), it is important to report such an outcome, if for no other reason than to minimise others wasting time and effort in pursing the same possibility.

I see no value in nitpicking minor and/or trivial issues and recommend the manuscript's publication as submitted.

Reviewer #2: This is an interesting paper that aimed at investigating whether maternal serum inhibin-A at 11-13 weeks could be as an alternative to PlGF or as an additional biomarker within the FMF screening test for preterm PE. In a nested case-control study, the authors found that replacing PlGF by inhibin-A or adding inhibin-A as an additional biomarker in and to the FMF triple screening test for preterm PE does not improve screening performance and will fail to identify pregnancies that are currently identified by the FMF triple test. It is a nicely conducted study and well written manuscript. I suggest modifying the following issues before the publication of the paper:

1) What can be the reason for lower inhibin-A levels in parous women with no previous history of PE? This could be shortly discussed.

2) Table 1: by looking at the differences between groups in many comparisons, several in the term PE vs control comparisons (e.g. IVF, nulliparous, etc) could be significant. Please double check the numbers.

3) It is stated that tests were considered statistically significant if the p-value was <0.05. What was the threshold after Bonferroni correction?

4) It is stated that “inhibin-A in the currently used FMF triple test does not significantly improve the screening performance could be explained by the concept of diminishing marginal returns and the fact that inhibin-A levels correlate with PlGF levels in PE groups.” However, there was no testing of any correlation. Please perform the test or remove the statement.

6. PLOS authors have the option to publish the peer review history of their article (what does this mean?). If published, this will include your full peer review and any attached files.

Reviewer #1: No

Reviewer #2: No

---

## [Author Response · Author response to Decision Letter 0]

25 Apr 2023

Response letter to reviewers’ and editors’ comment

Dear editors and reviewers,

Thank you very much for your comments and suggestion. The manuscript was amended, and the following points were addressed.

1) What can be the reason for lower inhibin-A levels in parous women with no previous history of PE? This could be shortly discussed

Response: Thank you, the discussion based on this issue was addressed as following: “We have further demonstrated that there are lower inhibin-A levels in parous women without prior PE, which reduces the risk for developing PE in the current pregnancy. This finding may indicate that women with a previous unaffected pregnancy have the ability to accomplish adequate maternal adaptation to tolerate pregnancy-related stress in the previous pregnancy, thus providing a protective effect against the development of PE in the current pregnancy..”

2) Table 1: by looking at the differences between groups in many comparisons, several in the term PE vs control comparisons (e.g., IVF, nulliparous, etc.) could be significant. Please double check the numbers.

Response: Thank you, the description of baseline characteristics was expanded as following: “Compared with the unaffected pregnancies, women with PE had higher maternal age and BMI, higher rates of IVF conception, nulliparity and previous history of PE, and lower rate of history of previous pregnancy without PE. There was no difference in the rates of smoking, chronic hypertension, and SLE/APS between groups. Although there was a higher rate of pre-existing diabetes mellitus in the PE group, compared to the unaffected group, the total number of cases was small.”

3) It is stated that tests were considered statistically significant if the p-value was <0.05. What was the threshold after Bonferroni correction?

Response: Thank you, ANOVA with Bonferroni correction with threshold (p value<0.016) for multiple comparison was used. It was added to the manuscript.

4) It is stated that “inhibin-A in the currently used FMF triple test does not significantly improve the screening performance could be explained by the concept of diminishing marginal returns and the fact that inhibin-A levels correlate with PlGF levels in PE groups.” However, there was no testing of any correlation. Please perform the test or remove the statement.

Response: Thank you, S2 Table which demonstrated observed log10 MoM biomarker distribution standard deviation and inter-biomarker correlations in women with and without preeclampsia was added. It demonstrated significant correlation between inhibin-A and PlGF levels.

Best regards,

Sakita Moungmaithong

---

## [Decision Letter · Decision Letter 1]

26 Jun 2023

Evaluation of first trimester maternal serum inhibin-A for preeclampsia screening

PONE-D-22-29184R1

Dear Dr. Poon,

We’re pleased to inform you that your manuscript has been judged scientifically suitable for publication and will be formally accepted for publication once it meets all outstanding technical requirements.

Kind regards,

Ahmed Mohamed Maged, MD

Academic Editor

PLOS ONE

Additional Editor Comments (optional):

All reviewers comments were clarified

Reviewers' comments:

Reviewer's Responses to Questions

**Comments to the Author**

1. If the authors have adequately addressed your comments raised in a previous round of review and you feel that this manuscript is now acceptable for publication, you may indicate that here to bypass the “Comments to the Author” section, enter your conflict of interest statement in the “Confidential to Editor” section, and submit your "Accept" recommendation.

Reviewer #1: All comments have been addressed

Reviewer #2: All comments have been addressed

2. Is the manuscript technically sound, and do the data support the conclusions?

Reviewer #1: Yes

Reviewer #2: Yes

3. Has the statistical analysis been performed appropriately and rigorously? 

Reviewer #1: Yes

Reviewer #2: Yes

4. Have the authors made all data underlying the findings in their manuscript fully available?

Reviewer #1: Yes

Reviewer #2: Yes

5. Is the manuscript presented in an intelligible fashion and written in standard English?

Reviewer #1: Yes

Reviewer #2: Yes

6. Review Comments to the Author

Reviewer #1: The authors have adequately addressed the minor corrections/amendments suggested by the review process.

Reviewer #2: The authors have revised their manuscript according to the comments of the reviewers, therefore, now it is acceptable for publication.

7. PLOS authors have the option to publish the peer review history of their article (what does this mean?). If published, this will include your full peer review and any attached files.

Reviewer #1: No

Reviewer #2: No

---

## [Editor Report · Acceptance letter]

2 Jul 2023

PONE-D-22-29184R1 

Evaluation of first trimester maternal serum inhibin-A for preeclampsia screening 

Dear Dr. Poon:

I'm pleased to inform you that your manuscript has been deemed suitable for publication in PLOS ONE. Congratulations! Your manuscript is now with our production department. 

Kind regards, 

on behalf of

Professor Ahmed Mohamed Maged 

Academic Editor

PLOS ONE